

# Identification of sterile a-motif domain-containing 9-like as a potential biomarker in patients with cutaneous melanoma

Junsen Ye[1],*, Haidan Tang[2],*, Chuanrui Xie[3], Wei Han[4],
Guoliang Shen[5], Ying Qian[5] and Jin Xu[4]

[1] The Department of Scientific Education, The First People's Hospital of Jiande, Hangzhou, China
[2] Affiliated Hospital of Youjiang Medical University for Nationalities, Guangxi, China
[3] The Department of Surgery, The People's Hospital of Rongcheng, Rongcheng, China
[4] Institute of Regenerative Biology and Medicine, Helmholtz Centre for Environmental Research–UFZ, Munich, Germany
[5] Department of Burn and Plastic Surgery, First Affiliated Hospital of Soochow University, Suzhou, China
* These authors contributed equally to this work.

Corresponding authors
Ying Qian, 18606270627@126.com
Jin Xu,
jin.xu@helmholtz-muenchen.de,
JinJyo@outlook.com

## ABSTRACT

Skin cutaneous melanoma (SKCM) is one of the most aggressive malignancies, accounting for approximately 75% of skin cancer-related fatalities annually. Sterile a-motif domain-containing 9-like (SAMD9L) has been found to regulate cell proliferation and suppress the neoplastic phenotype, but its specific role in SKCM remains unknown. To investigate the cancer-associated immunology of SKCM and the role of SAMD9L in tumor progression, we conducted an integrative bioinformatics analysis that revealed elevated expression levels of SAMD9L in SKCM. ROC curves and survival analyses confirmed the considerable diagnostic and prognostic abilities of SAMD9L. Moreover, a real-world cohort of 35 SKCM patients from the First Affiliated Hospital of Soochow University showed that higher expression levels of SAMD9L were associated with better prognosis. We performed validation experiments, including cell culture, generation of lentiviral-transfected SKCM cell lines, cell proliferation assay, and transwell assay, which demonstrated that down-regulation of SAMD9L significantly promoted proliferation and migration capacities of SKCM cells. Additionally, SAMD9L expression was found to be strongly linked to immune infiltration. Our results revealed a positive correlation between SAMD9L and XAF1 expression, suggesting that SAMD9L may serve as a prospective prognostic indicator of SKCM with co-expressed XAF1 gene.
In summary, our findings indicate that SAMD9L may serve as a promising prognostic and therapeutic biomarker and play a critical role in tumor-immune interactions in SKCM.

# INTRODUCTION

Skin cutaneous melanoma (SKCM) accounts for only 1% of all skin malignancies globally, but it is responsible for 75% of skin cancer-related deaths each year. With over 230,000 new occurrences and 100,000 new cases of invasive melanoma diagnosed annually,

melanoma is one of the most aggressive malignancies (*Schadendorf et al., 2018*). The Clark model outlines the progression of SKCM's pathogenesis, starting from melanocytes and advancing to dysplastic nevi, melanoma *in situ*, and ultimately invasive melanoma (*Clark et al., 1984*). While surgical resection is the primary treatment for early-stage tumors, advanced melanoma poses a challenge for chemotherapy and radiation therapy to be effective (*Gadeliya Goodson & Grossman, 2009*). Targeted therapies and immunotherapies have been effective in improving patient outcomes with metastatic melanoma (*Bajor et al., 2018*; *Flaherty, Hodi & Fisher, 2012*), including anti-PD-1/PDL-1 and anti-CTLA4 immunotherapy. However, only a small percentage of patients benefit from these treatments (*Braun, Burke & Van Allen, 2016*), necessitating the identification of biomarkers for rapid diagnosis and effective intervention.

Sterile α-motif domain-containing 9-like (SAMD9L) and its neighboring and close paralogue SAMD9 are located head-to-tail orientation in chromosome 7q21.2 in the human genome (*Ana Lemos de Matos, McFadden & Esteves, 2013*). Given their considerable sequence similarity, they may have comparable functions and be involved in related pathways. They participate in various biological processes, including an anti-proliferative effect and suppressing the neoplastic phenotype (*Li et al., 2007*). Deletions or deleterious mutations of SAMD9 and SAMD9L are usually found in human disorders, such as inherited acute myeloid leukemia and inflammatory calcified tumors (*Asou et al., 2009*; *Li et al., 2007*). However, SAMD9 mutations were only found in patients with normophosphatemic familial tumoral calcinosis, indicating that the two proteins may have different functions in humans (*Hershkovitz et al., 2011*). Breast cancer tissue expresses SAMD9L at a lower level than normal breast tissue (*Li et al., 2007*). Moreover, SAMD9L's capacity to hinder cell migration in activated human T cells could be linked to the activation of type I interferons (IFNα and β), underscoring the important role of SAMD9L in immune infiltration and human malignancies (*Pappas et al., 2009*).

Therefore, we explored the transcriptional and proteomics expression of SAMD9L and identified its prognostic value in SKCM patients. We hypothesized that SAMD9L may improve the prognosis of SKCM patients. Additionally, we investigated the role of SAMD9L in tumor-infiltrating immune cells (TIICs) and uncovered their interactions between the immune system and SKCM.

## METHODS

### Patients and variables

A total of 35 SKCM tissues were obtained from the First Affiliated Hospital of Soochow University (FAHSU), China, between 2010 and 2022. These patients had not received any prior radiotherapy or chemotherapy. The tissues were collected during surgery, fixed in 4% paraformaldehyde, and stored in the FAHSU tissue bank. Clinical information was obtained from hospital records. This study was approved by the Independent Ethics Committee of the FAHSU (IORG0008373). Written consent was obtained by all patients who were informed about the storage and use of their specimens for research purposes.

## Transcriptional expression of SAMD9L

The Oncomine database (http://www.oncomine.com) was used to examine the mRNA expression of SAMD9L in 20 common tumors (*Rhodes et al., 2004*). Using the following cut-off criteria, SAMD9L differential expression was chosen: $p = 0.01$ (Student's t-test), fold change = 1.5, and differentially expressed gene rank 10%.

Genotype-Tissue Expression project (GTEx; https://www.gtexportal.org/home/index.html) and the Cancer Genome Atlas (TCGA; https://tcga-data.nci.nih.gov/tcga/) provided data for customizable functionalities by Gene Expression Profiling Interactive Analysis (GEPIA, http://gepia.cancer-pku.cn/) (*Tang et al., 2017*). The transcriptional expression of SAMD9L was further investigated between SKCM and normal tissues by GEPIA. As for GSE15605, the transcriptional expression of SAMD9L was downloaded from GEO database (https://www.ncbi.nlm.nih.gov/geo/) and analyzed using T-test *via* GraphPad Prism (version 9.5.1).

## The Human Protein Atlas

The Human Protein Atlas (https://www.proteinatlas.org/) is a database with the aim to map all the human protein expression profiles in various cells, tissues, and organs (*Ponten, Jirstrom & Uhlen, 2008*). We extracted SAMD9L protein expression IHC images from clinical specimens of both normal skin tissues and SKCM patients using this database.

## Survival analysis

We used GEPIA to perform survival analysis, including establishing overall survival (OS) and disease-free survival (DFS), using log-rank test for the hypothesis evaluation. For the real-world cohort and GSE156030, survival analysis was performed by GraphPad Prism (version 9.5.1).

## Statistical analysis

From TCGA database, a whole population of 475 SKCM patients (104 primary and 371 metastatic samples) who have available RNA-sequence information along with corresponding clinical profiles were enrolled in this study. Both univariate and multivariate were performed using the Cox regression method. Therefore, independent variables, including age, gender, breslow depth, Clark level, pT stage, pN stage, pM stage, pathological stage, as well as SAMD9L expression was determined. Integrated score was identified as sum of the weight of SAMD9L as well as important clinicopathological prognostic indicators, including age, gender, Clark level, Breslow depth, TNM stage and pathologic stage. To compare the difference between different two groups, T tests were used. Statistically significant was defined as $p$-values < 0.05.

## Cell culture and generation of lentiviral-transfected SKCM cell lines

The human SKCM cell lines A375 and SK-MEL-28 were obtained from the Cell Bank of Shanghai Institutes of Biological Sciences, Chinese Academy of Sciences (Shanghai, China). The cells were cultured in RPMI 1640 medium (HyClone, Logan, Invitrogen, Carlsbad, CA, USA), containing 10% fetal bovine serum (FBS; Gibco, Grand Island, NY)
and 1% penicillin-streptomycin solution (Gibco, Grand Island, NY), at 37 °C in a humidified incubator with 5% $CO_2$. The medium was replaced every 3 days, and all tested cell lines were negative for mycoplasma using a Mycoplasma Detection kit (Lonza, Basel, Switzerland). Lentiviral inducible human siRNA from Dharmacon (ONTARGETplus SMARTpool, Dharmacon, Lafayette, CO, USA) was used to create the inducible SAMD9L knockdown cell lines, while non-targeting control siRNA was used as a negative control (NC).

## Western blot assay

The A375 and SK-MEL-28 cells were seeded at a density of $1 \times 10^5$ cells per 6-well plate. After transfection, the cells were washed with cold PBS, and total protein extracts were obtained by adding 80 μL of RIPA Lysis buffer. For Western blot analysis, 10 μg of protein from total lysates was loaded onto 8% polyacrylamide gels with 1× Laemmli buffer and separated by SDS-PAGE. The separated proteins were then transferred to an Immobilon-P PVDF membrane (IPVH00010; Millipore, Burlington, MA, USA), probed with SAMD9L Polyclonal Antibody (PA5-53994; Invitrogen, Thermo Fisher, Waltham, MA, USA) and GAPDH (MA1-16757; Invitrogen, Waltham, MA, USA).

## Cell proliferation assay

To perform CCK8 analysis, A375 and SK-MEL-28 cells were transfected and seeded into a 96-well plate at a density of 2,000 cells per well. The Cell Counting Kit-8 (CCK-8 kit; Dojindo, Japan) was added to the wells, and the proliferative capacity of the cells was determined following the manufacturer's instructions (*Wang et al., 2021*). The optical density (OD) values were measured using an automatic microplate reader (TEAN, Swiss) at 450 nm on days 1, 2, 3, 4, and 5 after seeding. Each sample was analyzed in triplicate.

## Transwell assay

To perform the cell migration assay, transwell chambers (Corning, Lowell, MA, USA) without matrigel coating were used. The cell density of different groups was adjusted to $1 \times 10^4$ cells/ml, and 100 μl of the cell suspension was added to the upper chamber. The lower 24-well plate chamber was filled with medium containing 20% fetal bovine serum. After 24 h, the negative control and transfected A375 and SK-MEL-28 cells that had migrated to the lower surface of the transwell chamber were treated with 4% polyoxymethylene for 15 min, deionized water, and 0.1% crystal violet for 30 min. Finally, the migrated A375 and SK-MEL-28 cells were counted in five random fields under an inverted microscope.

## Real-time quantitative PCR (RT-qPCR) analysis

Total RNA was extracted from 35 SKCM samples from the Soochow cohort using TRIzol® reagent (Invitrogen, Waltham, MA, USA). Primers were diluted in ddH$_2$O with SYBR Green PCR Master Mix (Applied Biosystems, Waltham, MA, USA). The fold change of SAMD9L relative to β-Actin was used to determine transcriptional expression. The PCR primers sequence for SAMD9L were as follows: forward 5′- AGT TCT TGA CCC CAA AGA AA-3′ and reverse 5′- CCT GGT GTC TCT CAG CCA GT-3′. SAMD9L mRNA expression was represented as ΔCt = Ct (SAMD9L) – ΔCt (β-actin).

## Genomic alteration of SAMD9L

SAMD9L mutation data in SKCM was obtained from cBioPortal database. The database is publicly available that allows users to search for multidimensional cancer genomics datasets (*Cerami et al., 2012*). We analyzed genomic alteration types, alteration frequency and overall survival. Genomic alterations of SAMD9L include copy number amplification, deep deletion, upregulation of mRNA, missense mutation with unknown significance, *etc.*

## Construction of protein–protein interaction (PPI) network

In this study, the Search Tool for the Retrieval of Interacting Genes (STRING, http://string-db.org) (version 11.0) (*Franceschini et al., 2013*) was utilized to measure the functional interactions between proteins and to characterize the co-regulation of hub genes at the protein level. The default threshold of interaction specificity score >0.4 in the STRING database was considered statistically significant.

## Functional annotations

Users of Database for Annotation, Visualization and Integrated Discovery (DAVID; http://david.ncifcrf.gov) (version 6.8) (*Sherman et al., 2022*) was applied to examine the biological significance of target genes using systematic and integrative functional annotation techniques. Therefore, functional annotations and pathway enrichment analysis was conducted by using DAVID, including the biological process (BP), cellular component (CC), and molecular function (MF), shown in a bubble chart. *p*-values lower than 0.05 were regarded as statistically significant.

Next, we utilized the bioinformatics software platform Cytoscape (version 3.7.2) to visualize molecular interaction networks. A Cytoscape plug-in called ClueGO (version 2.5.3) was used to explore GO and KEGG analyses based on target genes (*Bindea, Galon & Mlecnik, 2013*). Thus, GO:BP, CC, MF and KEGG functional enrichment were analyzed and plotted using ClueGO.

Based on the transcriptional sequences from the TCGA database, the related hallmarks were predicted using the Gene Set Enrichment Analysis (GSEA) method (*Subramanian et al., 2005*). In this study, GSEA was applied to classify genes based on their correlation with SAMD9L expression. The samples were divided into high- and low- SAMD9L groups using the median expression of SAMD9L. In each group, we conducted a Student's t-test on consistent pathways to identify differentially expressed genes, and calculated the mean of those genes. We then performed a permutation test with 1,000 iterations to detect hallmark pathways that were significantly involved. False positive results were corrected using the Benjamini and Hochberg (BH) method with a false discovery rate (FDR) of 0.25 or less. Genes with an adjusted *p*-value of less than 0.01 were considered significant.

## Immune infiltration analysis

The Tumor Immune Estimation Resource (TIMER) database (*Li et al., 2017*) is a comprehensive resource for users that can provide systematical analysis of immune infiltrates among diverse cancer types. TIMER was then utilized to examine the systematic correlation analysis between *SAMD9L* and TIICs signatures. Additionally, 28 different

kinds of TILs involved in interactions between immune system and tumors among human malignancies were identified using the integrated repository portal for tumor-immune system interactions (TISIDB, http://cis.hku.hk/TISIDB/index.php) (*Ru et al., 2019*). Based on the *SAMD9L* expression profile, gene set variation analysis was used to determine the relative abundance of TILs. The association between *SAMD9L* and TILs was calculated using the Spearman's test. *p*-values less than 0.05 were deemed statistically significant.

### Identification of SAMD9L co-expressed gene

The cBioPortal and GEPIA were used to find and validate the co-expressed genes of SAMD9L. Popular genomics browser UCSC Xena (http://xena.ucsc.edu/) offers visualization and integration for investigating and observing the public data hubs (*Lee et al., 2019*). With the application of data mining in TCGA SKCM through the UCSC Xena browser, the heat map along with correlation for both SAMD9L and XAF1 were produced.

## RESULTS

### Clinical and pathologic characteristics baseline of SKCM patients from TCGA and FAHSU cohorts

This study included 475 SKCM patients from the TCGA cohort and 35 from the FAHSU cohort. Table 1 presented the clinicopathological parameters of SKCM patients from both cohorts, including age at surgery, gender, location, Clark level, Breslow's depth, TNM stage, and pathologic stage.

### The differential expression of SAMD9L in various tumors

In light of the possibility that SAMD9L could serve as a crucial neo-target or neo-predictor for cancer diagnosis, we investigated expression of SAMD9L in different tumors and normal tissues through the Oncomine and GEPIA database to discover if *SAMD9L* expression is related to tumors. As Fig. 1A shows, the overexpression of *SAMD9L* was observed in many cancer types from the Oncomine database. In SKCM, the elevated *SAMD9L* expression was found compared with normal tissues, demonstrated in Fig. 1B. The expression profiling of *SAMD9L* indicated that there was an apparent heterogeneity in various tumors.

### The differential expression of SAMD9L in SKCM patients

We identified the expression of *SAMD9L* mRNA in both SKCM and normal tissues through GEPIA, which suggested that *SAMD9L* was significantly overexpressed in SKCM tissues compared to normal tissues ($p < 0.05$) (Fig. 2A). In addition, *SAMD9L* expression was also significantly higher in melanoma tissues than in normal skin tissues based on GSE15605 ($p = 0.0012$) (Fig. S1A). In accordance with *SAMD9L* mRNA expression, IHC staining showed that *SAMD9L* staining was low in normal skin tissues, while high levels of expression were seen in melanoma tissues (Fig. 2B). These findings indicated that overexpression of *SAMD9L* was observed at both transcriptional and proteomic levels in SKCM compared with normal tissues.

**Table 1 Clinicopathological characteristics of SKCM patients.**

| Characteristics | FAHSU cohort (N = 35) | TCGA cohort (N = 475) |
|---|---|---|
| *N* (%) | | |
| Age | | |
| ≤60 years | 13 (37.1) | 254 (54.7) |
| >60 years | 22 (62.9) | 211 (45.3) |
| Gender | | |
| Male | 16 (45.7) | 293 (61.9) |
| Female | 19 (54.3) | 180 (38.1) |
| Clark level | | |
| I | 17 (48.6) | 6 (1.9) |
| II | 15 (42.9) | 18 (5.6) |
| III–IV | 3 (8.5) | 246 (76.0) |
| V | 0 (0) | 53 (16.5) |
| Breslow depth (mm) | | |
| ≤0.75 | 6 (17.1) | 33 (9.4) |
| 0.76–1.50 | 12 (34.3) | 65 (18.6) |
| 1.51–4.00 | 13 (37.1) | 109 (31.1) |
| >4.00 | 4 (11.4) | 143 (40.9) |
| pT stage | | |
| T1–T2 | 23 (65.7) | 121 (33.2) |
| T3–T4 | 12 (34.3) | 244 (66.8) |
| pN stage | | |
| N0 | 35 (100) | 236 (64.4) |
| N1 | 0 (0) | 74 (20.2) |
| N2 | 0 (0) | 56 (15.4) |
| pM stage | | |
| M0 | 35 (100) | 419 (94.4) |
| M1 | 0 (0) | 25 (5.6) |
| Pathologic stage | | |
| I–II | 35 (100) | 232 (54.3) |
| III–IV | 0 (0) | 195 (45.7) |

Note:
SKCM, Skin Cutaneous Melanoma; FAHSU, The First Affiliated Hospital of Soochow University; TCGA, The Cancer Genome Atlas.

## Clinicopathological parameters associated with the expression of SAMD9L mRNA in SKCM patients

We discovered that *SAMD9L* expression was substantially linked with Breslow depth, Clark level, and T stage ($p < 0.05$) in SKCM samples after combining clinicopathological and expression profiles from TCGA. These three parameters all posed a decreased trend (Figs. S2A–S2C). While, the relationship between *SAMD9L* mRNA expression and M stage (Fig. S2D) may not have significant statistical significance.

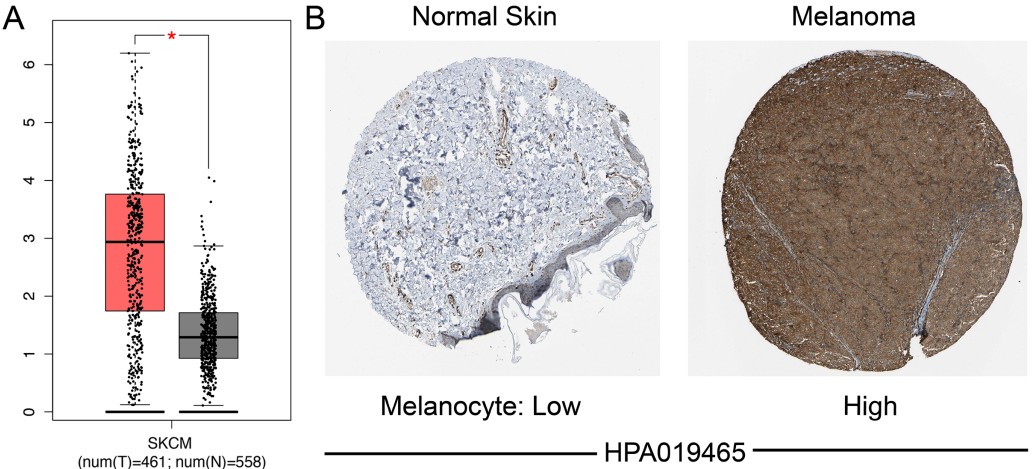

**Figure 1 The expression of *SAMD9L* in the cancerous tissues and in adjacent normal tissues.** The *SAMD9L* expression was analyzed in various cancerous tissues and normal tissues by (A) Oncomine and (B) GEPIA database. Also, the expression of *SAMD9L* was higher in primary SKCM compared to metastatic SKCM in TIMER database (C).               

**Figure 2 Differential *SAMD9L* expression in SKCM tissues and adjacent normal tissues.** (A) Transcriptional level of *SAMD9L* expression was found highly expressed in SKCM tissues compared with normal tissues by GEPIA (*$p < 0.05$). (B) Higher expression of *SAMD9L* is detected in melanoma tissues while low expression detected in normal tissues using the human protein atlas.

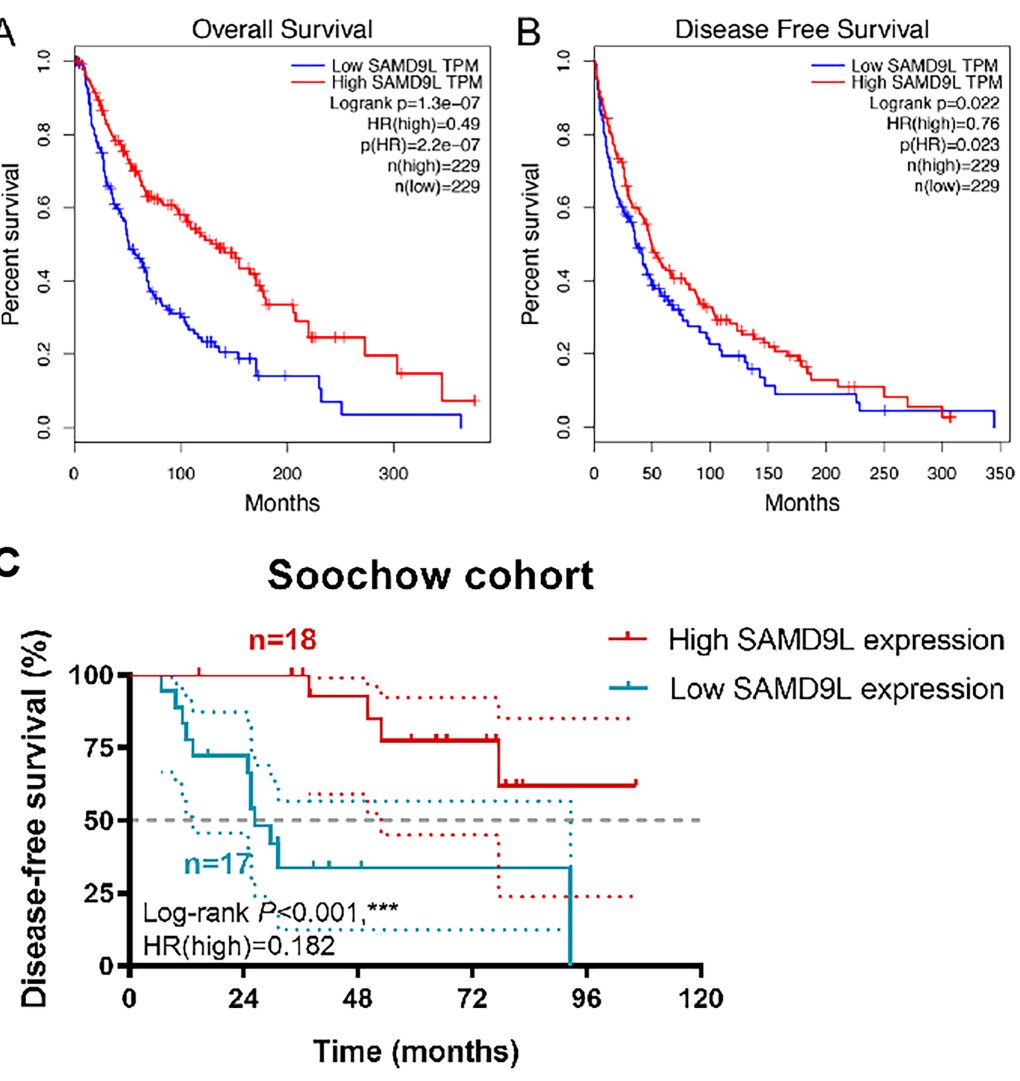

**Figure 3 Survival analysis.** Survival analysis in Kaplan–Meier method indicated that SAMD9L was significantly correlated with better (A) OS ($p$ = 1.31e-07) and (B) DFS ($p$ = 0.022) in SKCM patients. (C) Kaplan-Meier method showed that elevated SAMD9L expression was significantly correlated with longer DFS ($p$ < 0.001) in 35 SKCM patients from a real-world cohort.

## SAMD9L's potential predictive value in TCGA cohorts

Survival analysis in TCGA cohorts revealed a strong correlation between higher level *SAMD9L* expression and improved OS ($p$ < 0.001) and DFS ($p$ = 0.022) (Figs. 3A and 3B). Kaplan-Meier method showed that elevated *SAMD9L* expression was significantly correlated with longer DFS ($p$ < 0.001) in 35 SKCM patients from a real-world cohort (Fig. 3C). Survival analysis was also performed based on the GSE156030 in the Fig. S1B. Although the *p*-value was not significant ($p$ = 0.09), the survival analysis plot shows a consistent trend—higher expression of SAMD9L is associated with better prognosis.

Additionally, we established ROC curves to investigate the gene model's capacity to forecast prognostic incidents and created the formula: $1.02 \times$ Age + $2.02 \times$ pN stage +

**Table 2 Univariate and multivariate Cox regression analysis of OS in TCGA cohort.**

| Covariates | Univariate analysis | | | Multivariate analysis | | |
|---|---|---|---|---|---|---|
| | HR | 95% CI | p value | HR | 95% CI | p value |
| Age | 1.02 | [1.015–1.034] | 0.00 | 1.02 | [1.005–1.028] | 0.01 |
| Gender (ref. male) | 0.87 | [0.654–1.148] | 0.32 | – | – | – |
| Breslow depth | 1.03 | [1.015–1.041] | 0.00 | 1.01 | [0.997–1.030] | 0.11 |
| Clark level (ref. I-III) | 2.13 | [1.499–3.015] | 0.00 | 1.35 | [0.896–2.027] | 0.15 |
| pT stage (ref. T1-T2) | 1.99 | [1.508–2.619] | 0.00 | 1.30 | [0.89–1.884] | 0.18 |
| pN stage (ref. N0) | 1.67 | [1.267–2.197] | 0.00 | 2.02 | [1.095–3.710] | 0.02 |
| pM stage (ref. M0) | 1.89 | [1.052–3.399] | 0.03 | 2.58 | [1.181–5.644] | 0.02 |
| Pathological stage (ref. I-II) | 1.66 | [1.259–2.182] | 0.00 | 0.96 | [0.521–1.779] | 0.90 |
| SAMD9L expression (ref. low) | 0.50 | [0.386–0.659] | 0.00 | 0.63 | [0.450–0.868] | 0.01 |

**Table 3 Univariate and multivariate Cox regression analysis of RFS in TCGA cohort.**

| Covariates | Univariate analysis | | | Multivariate analysis | | |
|---|---|---|---|---|---|---|
| | HR | 95% CI | p value | HR | 95% CI | p value |
| Age | 1.02 | [1.003–1.031] | 0.02 | 1.005 | [0.987–1.024] | 0.559 |
| Gender (ref. male) | 0.66 | [0.413–1.064] | 0.09 | – | – | – |
| Breslow depth | 1.03 | [1.014–1.050] | 0.00 | 1.016 | [0.988–1.044] | 0.28 |
| Clark level (ref. I–III) | 1.87 | [1.054–3.303] | 0.03 | 0.995 | [0.52–1.904] | 0.989 |
| pT stage (ref. T1-T2) | 2.26 | [1.458–3.488] | 0.00 | 1.419 | [0.772–2.606] | 0.26 |
| pN stage (ref. N0) | 1.65 | [1.072–2.541] | 0.02 | 2.16 | [0.792–5.895] | 0.133 |
| pM stage (ref. M0) | 2.64 | [1.317–5.311] | 0.01 | 3.891 | [1.419–10.671] | 0.008 |
| Pathological stage (ref. I–II) | 1.68 | [1.097–2.586] | 0.02 | 1.182 | [0.434–3.221] | 0.744 |
| SAMD9L expression (ref. low) | 0.43 | [0.282–0.663] | 0.00 | 0.56 | [0.311–1.010] | 0.56 |

2.58 × pM stage + 0.63 × SAMD9L expression for OS after including all the significant clinicopathological factors and gene expression profiles in the Cox regression models (Tables 2 and 3). The AUC index in the OS and RFS were 0.704 and 0.716, respectively ($p < 0.001$; Figs. S3A–S3B).

## Down-regulation of SAMD9L promotes proliferation and migration of A375 and SK-MEL-28 cells

To reveal malignant behaviors of *SAMD9L in vitro*, we verified the down-regulated of *SAMD9L* in the siRNA-transfected group compared with negative control group in A375 and SK-MEL-28 cells (Fig. 4A). The results of the CCK-8 assay showed that the downregulation of SAMD9L significantly enhanced the proliferative ability of SKCM cells compared to the control group (Fig. 4B). Additionally, the transwell migration assay revealed that the downregulation of SAMD9L significantly promoted the metastatic ability of SKCM cells (Figs. 4C–4D). Collectively, these findings demonstrated that the

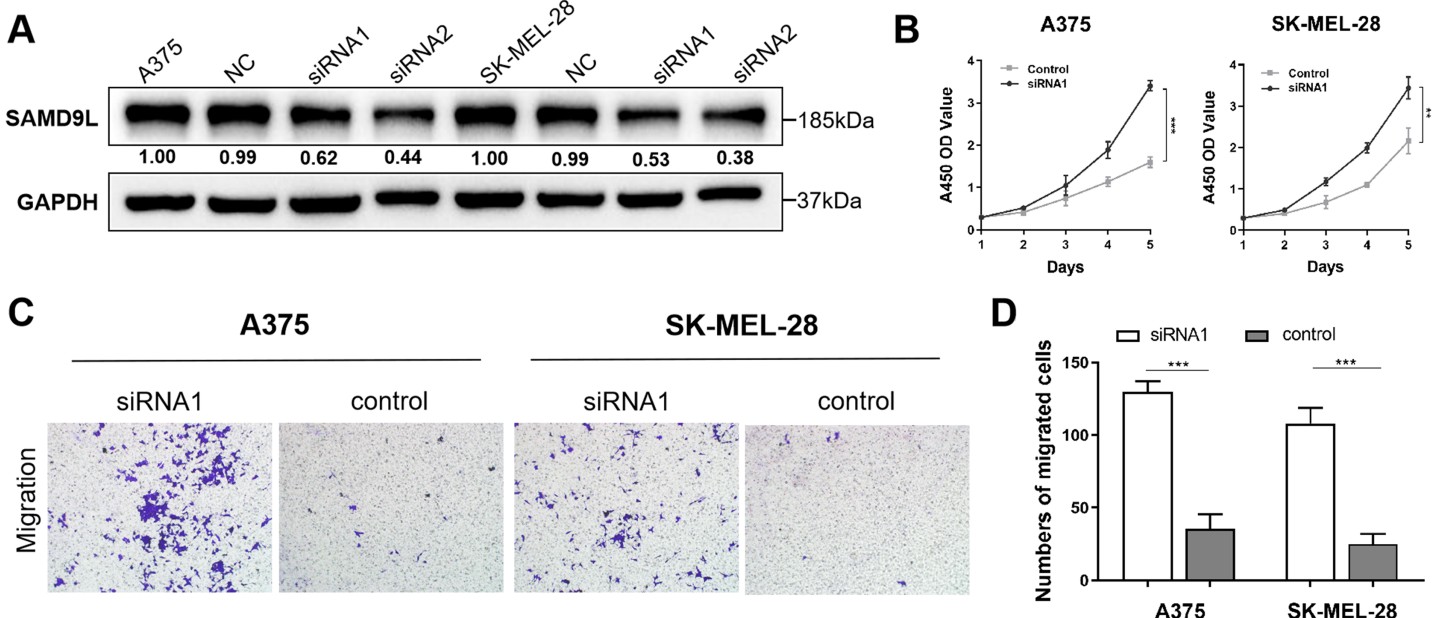

**Figure 4 Validation of SAMD9L in melanoma.** (A) To reveal malignant behaviors of SAMD9L *in vitro*, we explored and confirmed the down-regulated of SAMD9L in the siRNA-transfected group compared with negative control group in A375 and SK-MEL-28 cells. (B) According to the results of the CCK-8 assay, down-regulated expression of SAMD9L expression significantly increased the proliferative ability of SKCM cells compared with control group. (C and D) Transwell migration assay indicated that the down-regulation expression of SAMD9L expression markedly promoted the metastasis ability of SKCM cells.

downregulation of SAMD9L significantly enhanced the proliferation and migration capacities of SKCM cells.

## Genomic alteration of SAMD9L

By using the cBioPortal web server at SKCM, we conducted further research on the *SAMD9L* alteration status, including a total of 448 patient samples from the TCGA SKCM database for analysis. In *SAMD9L*, the frequency of changes was 23%. We found that missense mutations were the most common form of alteration, followed by mRNA high, truncating mutation, and amplification (Figs. S4A–S4B). Overall survival analysis between altered and unaltered group revealed that the altered group was correlated with better prognosis ($p = 0.0478$) (Fig. S4C).

## Functional annotations of SAMD9L

Figure 5A depicted an array of the co-expression genes for *SAMD9L*. Functional enrichment analysis was carried out among the 11 implicated genes and the results were then visualized in a bubble chart (Fig. 5B). Positive genes were significantly involved in both defense response to virus and response to virus, type I interferon signaling pathway, cellular response to interferon-alpha, negative regulation of protein binding, cytoplasm and RNA binding. As shown in Fig. 5C, the results of functional annotation using ClueGO were similar with the bubble chart, indicating *SAMD9L* were significantly associated with response to virus, defense response to virus, response to type I interferon, cellular response to type I interferon and type I interferon signaling pathway.

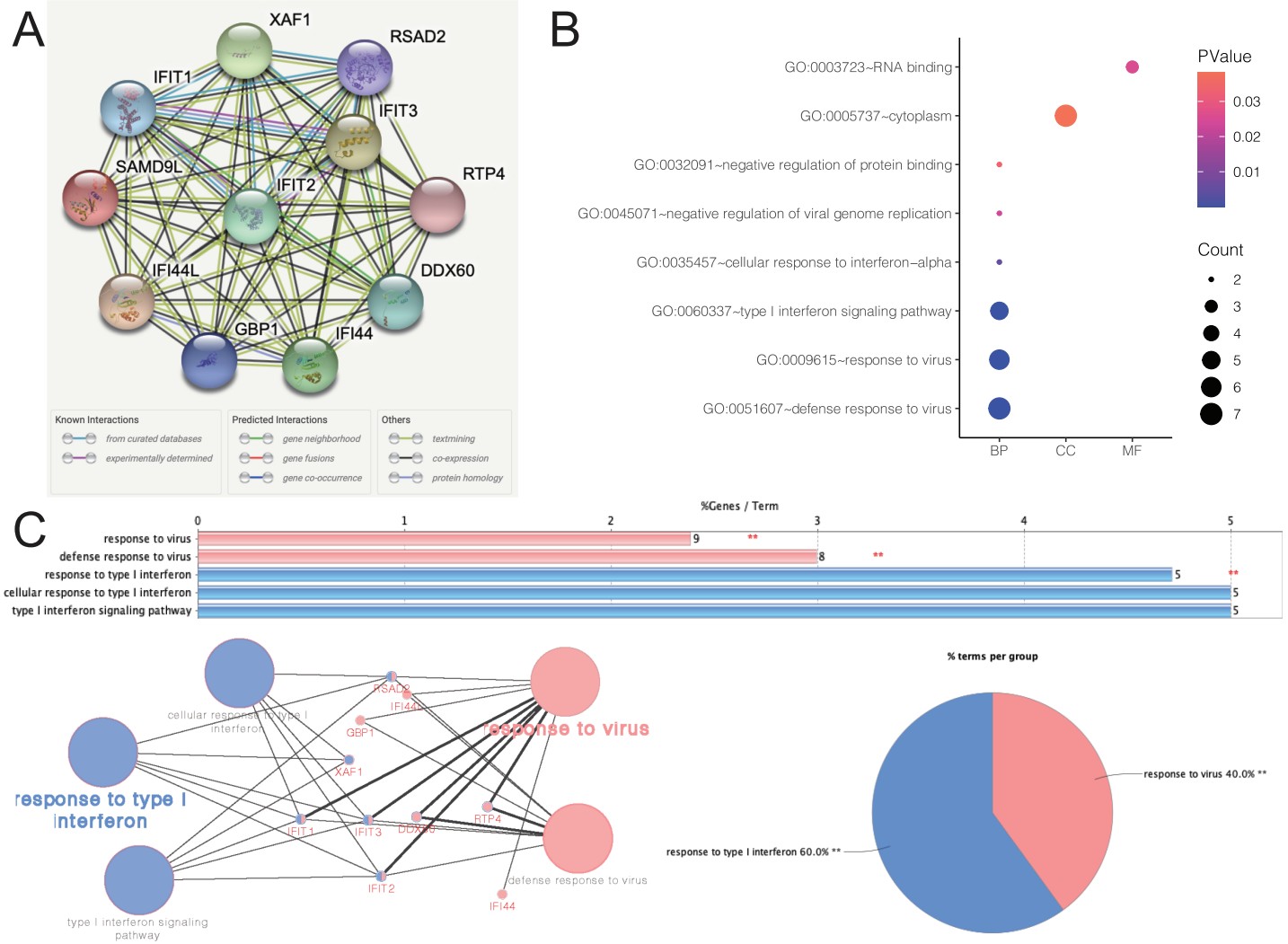

**Figure 5 Functional annotations and predicted signaling pathways.** (A) The PPI network of *SAMD9L* was constructed. A network of *SAMD9L* and its co-expression genes was set up visually. (B) Functional enrichment analyses of a total of 11 involved genes were performed and visualized in bubble chart. (C) Functional annotation using ClueGO indicated that *SAMD9L* were significantly associated with response to virus, defense response to virus, response to type I interferon, cellular response to type I interferon and type I interferon signaling pathway.

## Related significant genes and pathways

The *SAMD9L* hallmark analysis was performed using GSEA analysis. According to the results, such as allograft rejection, inflammatory response, complement, apoptosis, coagulation, interferon alpha response, interferon gamma response, Kras signaling, IL2–STAT5 signaling, IL6/JAK–STAT3 signaling and TNF-A signaling *via* NF-κB are most relevant pathways, shown in detail in Figs. 6A–6G.

## Correlation of SAMD9L and immune infiltration level

To investigate the effect of *SAMD9L* expression on SKCM in tumor microenvironment (TME), we found significant correlations of *SAMD9L* with 28 types of TILs among human heterogeneous cancers (Fig. 7A). As Figs. 7B–7G showed, *SAMD9L* played a critical role in

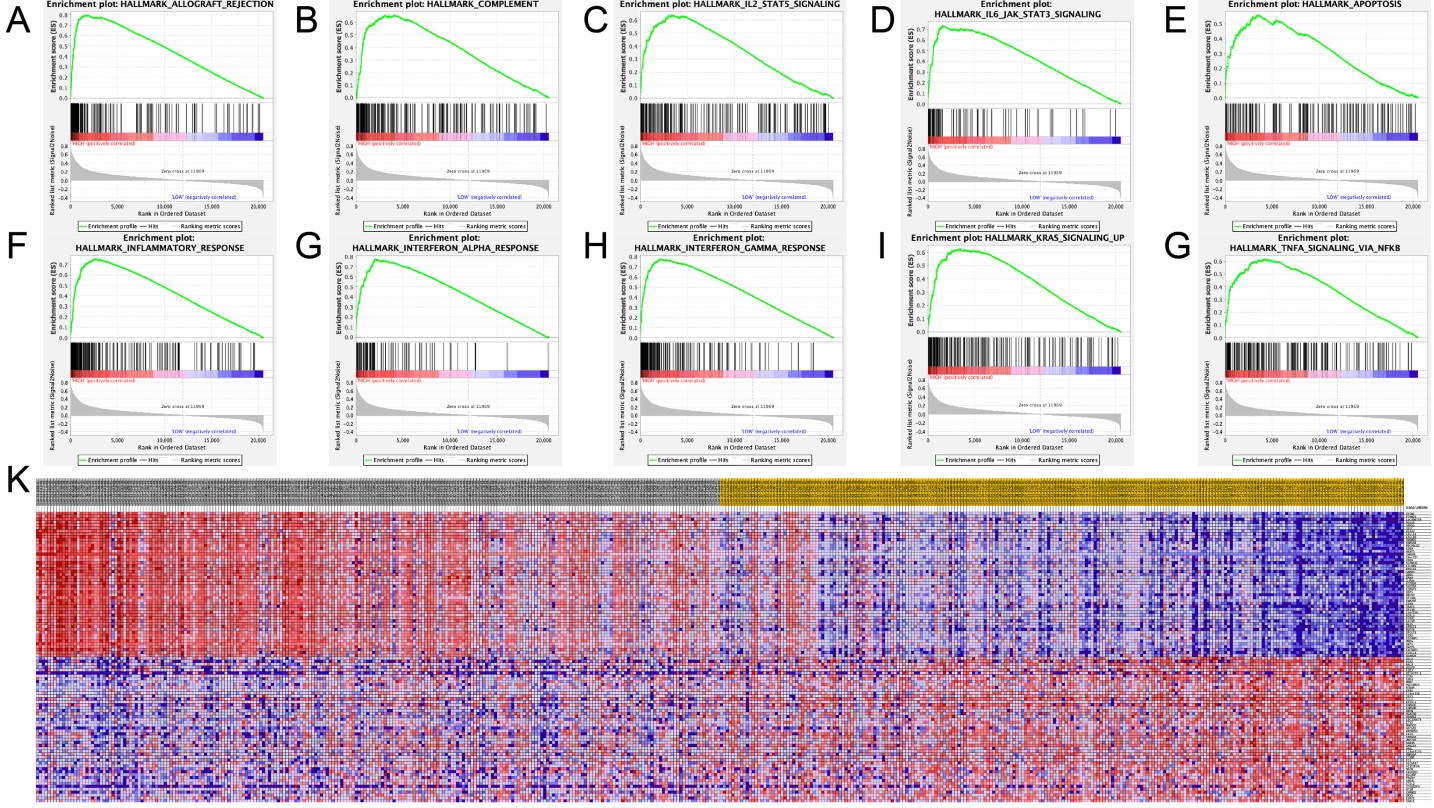

**Figure 6 Significant related genes and hallmarks pathways in SKCM obtained by GSEA.** (A–J) The most involved significant pathways included allograft rejection, complement, coagulation, IL2–STAT5 signaling, IL6/JAK–STAT3 signaling, apoptosis, inflammatory response, interferon alpha response, interferon gamma response, Kras signaling and TNF-A signaling *via* NF-κB.

immune infiltration that significantly correlated with abundance of natural killer cells (NK cells; rho = 0.526, $p < 0.001$), natural killer T cells (NK T cells; rho = 0.539, $p < 0.001$), myeloid derived suppressor cells (MDSC; rho = 0.513, $p < 0.001$), activated dendritic cells (act DC, rho = 0.451, $p < 0.001$), eosinophil (rho = 0.408, $p < 0.001$), and mast cells (rho = 0.407, $p < 0.001$). Furthermore, to thoroughly investigate the molecular characteristics of tumor-immune interactions. We performed TIMER analysis.

As depicted in Fig. 7H, TIMER analysis revealed significant favorable relationship with infiltrating levels of B cell ($r = 0.182$, $p = 1.05\text{e}{-}04$), CD8+ T cells ($r = 0.549$, $p = 7.12\text{e}{-}36$), CD4+ T cells ($r = 0.335$, $p = 3.53\text{e}{-}13$), macrophages ($r = 0.307$, $p = 2.35\text{e}{-}11$), neutrophils ($r = 0.742$, $p = 5.01\text{e}{-}80$) and dendritic cell ($r = 0.535$, $p = 1.90\text{e}{-}34$) in SKCM. Particularly, a positive correlation between *SAMD9L* CNV and infiltrating levels of B cells, CD4+ T cells, and dendritic cells was observed (Fig. 7I).

## Co-expression of SAMD9L gene

Ultimately, we used the cBioportal database to analyze the co-expression of the *SAMD9L* gene (Fig. 8A). Targeting protein for X-linked inhibitor of apoptosis associated factor 1 (*XAF1*) is the top correlated gene, which is an inhibitor of apoptosis. A positive correlation between *SAMD9L* and *XAF1* expression was revealed (Fig. 8D). Moreover, a closely

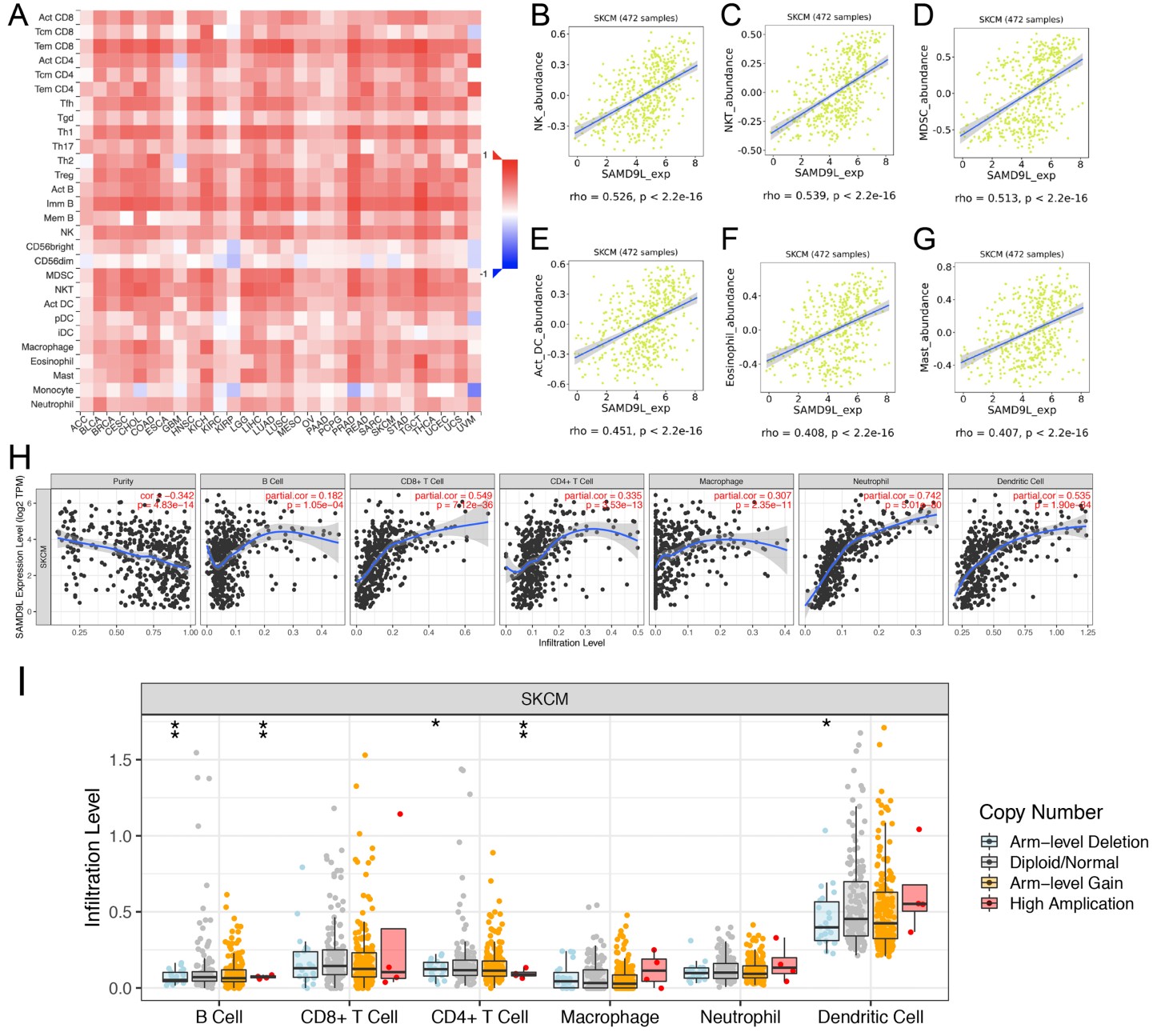

**Figure 7 Correlations between expression of *SAMD9L* and immune infiltration level.** (A) Relations between expression of *SAMD9L* and 28 types of TILs across human cancers. (B–G) *SAMD9L* significantly correlated with abundance of natural killer cells (NK cells; rho = 0.526, *p* < 0.001), natural killer T cells (NK T cells; rho = 0.539, *p* < 0.001), myeloid derived suppressor cells (MDSC; rho = 0.513, *p* < 0.001), activated dendritic cells (act DC, rho = 0.451, *p* < 0.001), eosinophil (rho = 0.408, *p* < 0.001), and mast cells (rho = 0.407, *p* < 0.001). (H) TIMER analysis demonstrated significant positive associations with infiltrating levels of B cell (*r* = 0.182, *p* = 1.05e−04), CD8+ T cells (*r* = 0.549, *p* = 7.12e−36), CD4+ T cells (*r* = 0.335, *p* = 3.53e−13), macrophages (*r* = 0.307, *p* = 2.35e−11), neutrophils (*r* = 0.742, *p* = 5.01e−80) and dendritic cell (*r* = 0.535, *p* = 1.90e−34) in SKCM. (I) *SAMD9L* CNV has significant correlations with infiltrating levels of B cells, CD4+ T cells, and dendritic cells. $0 \leq$ *** $< 0.001 \leq$ ** $< 0.01 \leq$ * $< 0.05$.

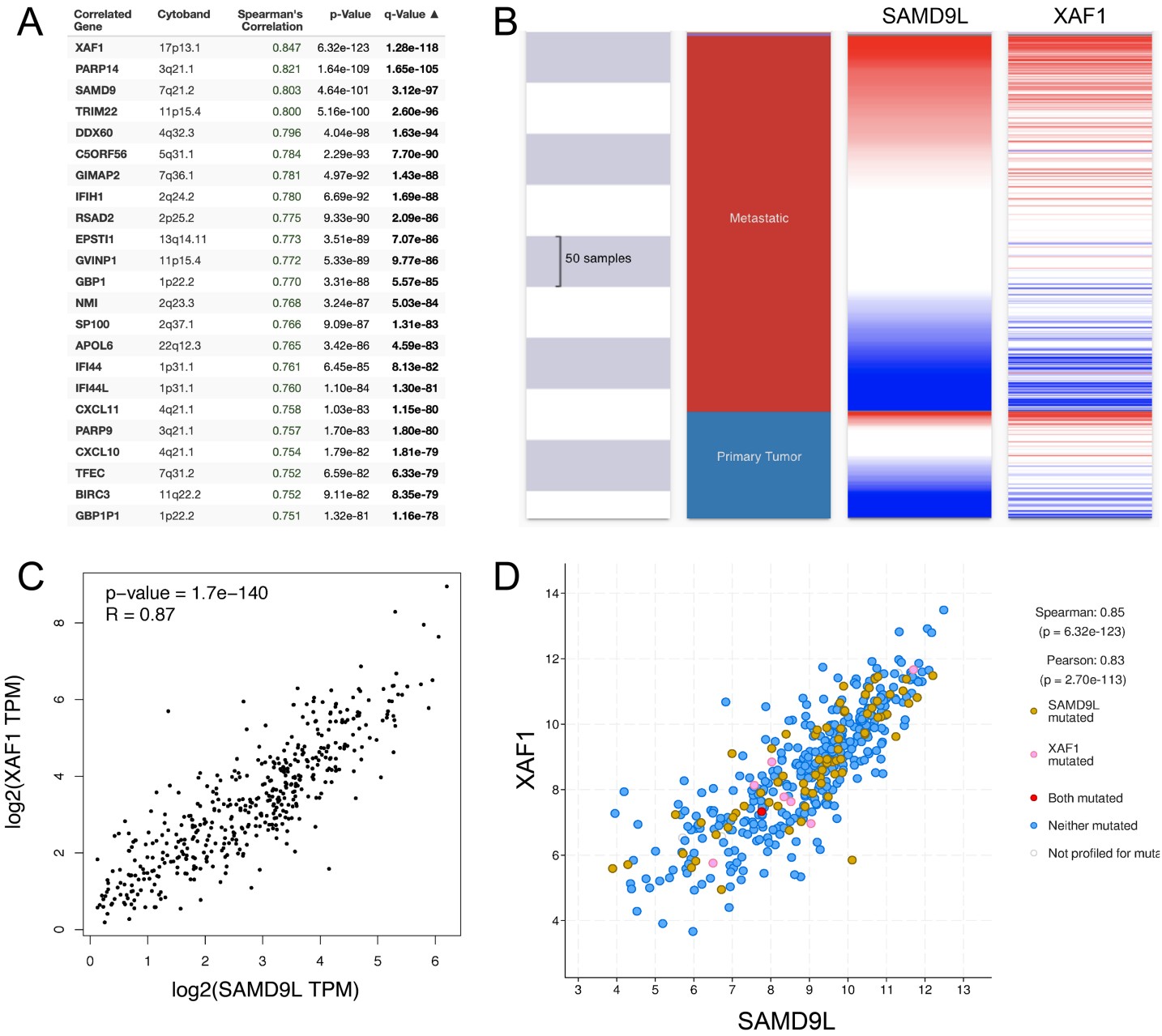

**Figure 8  Co-expression of SAMD9L gene.** (A) The co-expression genes of SAMD9L using the cBioportal database. Targeting protein for XAF1 is the top correlated gene, which is an inhibitor of apoptosis. A positive correlation between SAMD9L and XAF1 expression was revealed. (B–D) Consistent with the results of cBioPortal, we used the UCSC Xena web-based tool and GEPIA, and confirmed a positive correlation between SAMD9L and XAF1 expression, as shown in the heat map. These results suggested that SAMD9L might be closely related to the XAF1 signaling pathway in SKCM.

correlation between *SAMD9L* and *XAF1* expression was also validated after verifying SKCM patient data in the TCGA database using the UCSC Xena web-based tool and GEPIA, as illustrated in the heat map (Figs. 8B–8C). These findings showed that *SAMD9L* and the *XAF1* signaling pathway in SKCM could be closely connected.

## DISCUSSION

The malignant transition of melanocytes to metastatic melanoma involves a complex interaction of several exogenous and endogenous variables, including tumor-intrinsic and immune-related factors. Genetic alterations stimulate the phosphoinositide-3-kinase (PI3K), phosphatase-and-tensin homologue (PTEN), mitogen-activated-protein-kinase (MAPK), protein-kinase-B (AKT), and mammalian target of rapamycin (mTOR) pathways in a reciprocal manner (*Schadendorf et al., 2018*). Immune checkpoints serve as "on" and "off" regulators for immune cells. When switched to "on", immune cells like T cells can detect infection and inflammation in the body and recognize abnormal and cancer cells for eradication. Melanoma cells can circumvent the immune system by strengthening immune checkpoints to prevent escalating immune responses. *SAMD9L* has been recognized as a tumor suppressor in the development and progression of tumors such as myeloid malignancy (*Asou et al., 2009*) and HBV-associated HCC (*Wang et al., 2014*), but its relationship with melanoma remained unclear.

Our study is the first to delve into the role of *SAMD9L* in melanoma. We utilized the TCGA and GEO database to retrieve SAMD9L gene mRNA expression and SKCM patients' survival data and evaluated the underlying use of differential SAMD9L gene mRNA expression as prognostic indicators. We also recruited 35 SKCM patients from a real-world cohort. Our results showed increased expression of SAMD9L on both mRNA and proteomic levels in SKCM. Validation experiments confirmed the down-regulation of SAMD9L significantly promoted proliferation and migration capacities of SKCM cells. Moreover, based on K-M methods and COX models, elevated expression of SAMD9L was associated with better survival outcomes, indicating its significant role in SKCM patients' prognosis. According to the DFS analysis from our real-world cohort, it was consistent with survival analysis from the TCGA cohort. These findings suggest that SAMD9L could serve as a potential biomarker for the diagnosis and treatment of SKCM, particularly as a prognostic indicator. Our study also sheds light on the relationship between *SAMD9L* and the tumor-infiltrating immune cells (TIICs) and their interactions with the immune system in SKCM.

Next, we conducted functional enrichment analysis to investigate the co-expression of genes with SAMD9L and analyzed its hallmarks. Our analysis revealed that SAMD9L plays a crucial role in several significant hallmarks pathways in SKCM samples, including allograft rejection, inflammatory response, complement, apoptosis, coagulation, interferon alpha response, interferon gamma response, Kras signaling, IL2–STAT5 signaling, IL6/JAK–STAT3 signaling, and TNF-A signaling *via* NF-κB.

Cancer-related inflammation (CRI) is a recognized hallmark of cancer that contributes to the development and progression of malignancies (*Colotta et al., 2009*; *Crusz & Balkwill, 2015*; *Hanahan & Weinberg, 2011*) which can predict melanoma patients' response to blockade of immune checkpoint (*Holzel & Tuting, 2016*). CRI is also associated with correlative genetic alterations that contribute to the progression and metastasis of tumors (*Singh, Mishra & Aggarwal, 2017*). *SAMD9L* in SKCM regulates multiple

inflammation-related signaling pathways, including IL2–STAT5 signaling, IL6/JAK–STAT3 signaling, and TNF-alpha signaling pathways.

IL-2 has been proven to be an effective therapy for metastatic melanoma patients as it can stimulate cytotoxic T lymphocytes (*Fang et al., 2019*). Infusion of IL-2 in treating metastatic melanoma patients was considered as the first success in tumor immunotherapy (*Sim & Radvanyi, 2014*). Immune-related anti-tumor actions could be influenced by the IL2-STAT5 signaling pathway. The inhibition of the STAT5 signaling pathway by IL-2 could be an effective immunotherapy for SKCM patients.

Abnormal hyperactivation of the IL-6/JAK–STAT3 pathway is associated with poor prognosis in different types of cancer (*Chen et al., 2013*; *Johnson, O'Keefe & Grandis, 2018*; *Macha et al., 2011*). The phosphorylation of STAT3 activated by IL-6 leads to the transcription of genes that regulate tumor cell proliferation and anti-apoptosis (*Na et al., 2013*). Additionally, IL-6 promotes melanoma cell invasion by driving its ability to metastasize (*Kushiro et al., 2012*). Melanoma cells express JAK-STAT-mediated PD-1 ligands PD-L1 and PD-L2, which can be triggered by released interferons after tumor antigen recognition by T cells (*Johnson, O'Keefe & Grandis, 2018*; *Schadendorf et al., 2018*).

TNF and its family members play a dual role in regulating immune responses, tumorigenesis, viral replication, and immune diseases (*Aggarwal, 2003*). In the context of T cell immunotherapy, melanoma cells that are exposed to the proinflammatory cytokine TNF-α may generate reversible dedifferentiation. Moreover, these TNF-α-mediated tumor cells are poorly recognized by T cells specific for melanocytic antigens (*Landsberg et al., 2012*). Fas signaling, which is a non-apoptotic TNF-family receptor, reduces the effectiveness of T cell adoptive immunotherapy and regulates increased motility and invasiveness of tumor cells (*Barnhart et al., 2004*). NF-κB transcription factors are critical modulators of the immune system and are essential for acute immune processes during inflammation. Interestingly, NF-κB can be activated by Fas under specific circumstances to protect against foreign pathogens and malignancies (*Karin & Greten, 2005*; *Oeckinghaus, Hayden & Ghosh, 2011*). Our GSEA analysis revealed a strong correlation between *SAMD9L* and TNF signaling and the NF-κB pathway, suggesting that *SAMD9L* could serve as a promising target for anti-tumor immunity.

Interferons (IFNs) are a family of cytokines that play a crucial role in the immune response against viral infections and tumors. Recent studies have shown that IFNs are also important in the generation of an anti-tumor immune response (*Benci et al., 2016*). In melanoma, the activity of the IFN pathway can predict the response to MEK inhibition (*Litvin et al., 2015*). Adjuvant IFN-α therapy has been widely used in the clinical treatment of melanoma, and when combined with other immunotherapies, it can provide a synergistic effect (*Di Trolio et al., 2015*; *Vihinen et al., 2015*). However, it has been observed that IFN-γ-induced upregulation of immunological checkpoints, such as PD-1 and PD-L1, may contribute to an immunosuppressive and tolerogenic tumor microenvironment (*Mo et al., 2018*; *Schreiber, Old & Smyth, 2011*). In our study, we found a close correlation between the expression of SAMD9L and the IFN-α and IFN-γ responses in patients with SKCM.

In addition, we unraveled that *SAMD9L* expression is strongly linked with *XAF1* using cBioPortal and UCSC Xena web-based tools. *XAF1* is a 301-amino-acid nuclear protein containing seven zinc fingers, known as the best characterized and the most potent of apoptosis (*Kin Cheung et al., 2004*). Previous study revealed that melanoma's expression of *XAF1* was obviously lower than nevus tissues (*Kin Cheung et al., 2004*). *Leaman et al. (2002)* identified *XAF1* as a novel IFN-stimulated gene, which can strongly influence the cellular sensitivity to the proapoptotic actions of tumor necrosis factor-related apoptosis-inducing ligand, including melanoma. These findings, along with our research of *SAMD9L*, shed light on that *SAMD9L* gene together with *XAF1* expression might inhibit the progression of tumor.

However, several points need to be addressed despite our rigorous data analysis in this study. Firstly, in the multivariate Cox regression analysis of RFS in the TCGA cohort, SAMD9L expression was not significantly correlated, which may be due to the confounding effects of other clinical factors on RFS. Secondly, it should be noted that the AUC values under the ROC curves were below 0.8, indicating that further evidence is needed to confirm the robustness of our findings. Thirdly, larger cohorts and *in vivo* experiments are needed to verify the absoluteness of these findings.

## CONCLUSION

In summary, our study is the first to elucidate the correlation of SAMD9L and melanoma comprehensively. The functional network of *SAMD9L* was correlated to the IL2–STAT5 signaling, IL6/JAK–STAT3 signaling, apoptosis, inflammatory response, interferon alpha response, interferon gamma response, Kras signaling and TNF- alpha signaling *via* NF-κB. Furthermore, the expression of *SAMD9L* was found to be highly associated with immune infiltration. Therefore, these results shed light on the association between *SAMD9L* and tumor-immune interactions. *SAMD9L* might be presented as a new valuable prognostic and therapeutic indicator in SKCM.

### Funding
The authors received no funding for this work.

### Competing Interests
The authors declare that they have no competing interests.

### Author Contributions
- Junsen Ye conceived and designed the experiments, performed the experiments, analyzed the data, prepared figures and/or tables, and approved the final draft.
- Haidan Tang conceived and designed the experiments, performed the experiments, analyzed the data, prepared figures and/or tables, and approved the final draft.
- Chuanrui Xie conceived and designed the experiments, performed the experiments, analyzed the data, prepared figures and/or tables, and approved the final draft.

- Wei Han conceived and designed the experiments, performed the experiments, analyzed the data, prepared figures and/or tables, collect the patients information and follow up with the patient, and approved the final draft.
- Guoliang Shen conceived and designed the experiments, performed the experiments, prepared figures and/or tables, collect the patients information and follow up with the patient, and approved the final draft.
- Ying Qian conceived and designed the experiments, performed the experiments, analyzed the data, authored or reviewed drafts of the article, collect the patients information and follow up with the patients, and approved the final draft.
- Jin Xu conceived and designed the experiments, performed the experiments, analyzed the data, authored or reviewed drafts of the article, and approved the final draft.

### Human Ethics

The following information was supplied relating to ethical approvals (*i.e.*, approving body and any reference numbers):

The First Affiliated Hospital of Soochow University granted ethical approval to carry out the study within its facilities (IORG0008373).

### Data Availability

The raw data is available in the Supplemental Files.

### Supplemental Information

Supplemental information for this article can be found online at http://dx.doi.org/10.7717/peerj.15634#supplemental-information.

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
