# Peer review of "Identification of sterile a-motif domain-containing 9-like as a potential biomarker in patients with cutaneous melanoma"

_PeerJ, doi:10.7717/peerj.15634_

## Round 0.1 · original submission · Major Revisions

In addition to the points raised by the reviewers, Language editing is needed.

Reviewer 1 ·

Basic reporting

The manuscript documents the role of SAMD9L in tumor progression and cancer-associated immunology of SKCM detected by integrative bioinformatics and subsequent experimental analyses. The case aims to bring attention to a previously unreported molecular alteration of SAMD9L and suggests that indicate SAMD9L may serve as an underlying promising prognostic and therapeutic biomarker and play a critical role in tumor-immune interactions in SKCM. The clinical relevance of this findings was also validated in this research which is very meaningful.

Experimental design

1. In the method part, "Additionally, 28 different kinds of TILs of interactions of immune system and tumor among human malignancies were identified”. Please note that TIL only includes two types. Immune cells would be more appropriate here.

Validity of the findings

2. In the result part, “Additionally, we established ROC curves to investigate the gene model's
capacity to forecast prognostic incidents and created the formula: 1.02×Age+2.02×pN stage +2.58×pM stage + 0.63×SAMD9L expression”. Please explain why T stage and gender were not included.

Additional comments

3. Some typos/grammatical errors were noticed. Another round of careful editing to make these all consistent will clean up the manuscript.
4. The reasons for selecting SAMD9L for further analysis should be clarified.
5. Some figures with few information could been merged or removed to Supplements to decrease the number of figures.

·

Basic reporting

The authors of this article focused on the effect of SAMD9L on the progression and prognosis of SKCM patients, starting with bioinformatic analysis and validation of the effect of AAMD9L on SKCM through TCGA and GTEx datasets. Secondly, the relevant results were validated and further analyzed using data from 35 SKCM patients collected in their hospital. The article is clear, statistically correct, and overall a good article.
The English language should be improved to ensure that an international audience can clearly understand your text. The current phrasing makes comprehension difficult. I suggest you have a colleague who is proficient in English and familiar with the subject matter review your manuscript, or contact a professional editing service.

Experimental design

The experiments are well-designed and include bioinformatics analysis, clinical data validation, and cytological experimental exploration.

Validity of the findings

A total of 35 patients were validated for clinical data, and the smaller sample size may affect the stability of the results. If possible, could the authors perform a validation of the results of the TCGA and GTEx database analysis in the GEO database. As far as I know, there is a lot of clinical and transcriptomic data on SKCM patients in the GEO database.

Additional comments

1. In the multivariate Cox regression analysis of RFS in the TCGA cohort, SAMD9L expression was not significantly correlated, and the authors should elucidate the reasons for the correlation in the Discussion.
2. Figure 4 Why OS results were not provided in the analysis of the Soochow dataset.
3. The AUC values under the ROC curves in Figure 5 were not higher than 0.8, suggesting that the individual predictive potency of this gene was not robust.
4. Compared to TIMER, the results of CibersortX may be more accurate, and the authors may try it later.

Reviewer 3 ·

Basic reporting

In this work, the authors have assessed SAMD9L as a potential biomarker in tumor progression and cancer-associated immunology of melanoma, using integrative bioinformatics approaches. They found the elevated expression of SAMD9L in melanoma, which was closely associated with the immune-infiltrating levels of CD4+ T, CD8+ T, neutrophils, macrophages and dendritic cells.

Experimental design

Overall, the study demonstrates that high SAMD9L levels in melanoma may be attributable to T cell infiltration which lead to improved outcomes in melanoma patients, and thus may serve as a promising prognostic and therapeutic biomarker in melanoma.

Validity of the findings

The data are interesting while I have some concerns.

Additional comments

1. "Higher expression of SAMD9L is detected in melanoma tissues while low expression detected in normal tissues using the Human Protein Atlas " To support these data, quantification of SAMD9L expression at the protein level should be well described and shown, instead of merely displaying IHC images.
2. As presented in the ‘Discussion’ section, SAMD9L is recognized as a tumor suppressor “lines 248-249”. However, the finding of this study is different than the known role of SAMD9L in others cancer. The authors are suggested to elaborate on the differential role of SAMD9L in melanoma.
3. The language of the whole paper needs to be improved. At first glance at your manuscript, I noticed several grammar errors. Please check the WHOLE paper carefully or have it professionally edited.
4. How the GSEA analysis was performed was unclear from the description provided.
5. The references are not presented in the same style. Multiple references are inadequately presented with missing element(s).

·

Basic reporting

The authors has very clearly explained the importance of SAMD9L (which is a known tumor suppressor) in SKCM. The writing is clear with a little correction from my side:
1) In Introduction section line number 63-65 (Furthermore.....human malignancies) can be rewritten to make more sense. It is not very clear.
2) In material methods : The line 106-107( Integrated score...prognostic indicators) seems vague. To make it more clear I suggest that authors might try to give an example of the important clinico-pathological prognostic indicators that they used for the sum.
3) I am not able to understand why in line number 101 and 198 its written that 475 TCGA patients were included for analysis while line number 247 talks only about 444 patients. were some of the patient samples were excluded if yes then what was the criteria?
4) In line number 284 XAF1 is explained as X-linked inhibitor of apoptosis, rather according to my knowledge it should be addressed as X-linked inhibitor of apoptosis associated factor.
The article is really interesting and i appreciate authors for all the analysis and data generation.

Experimental design

The experiment design is very well organised and performed.
I have only minimal concern regarding the material and methods :
1) For transwell assay : In line 135-136 : Why the chambers were used without the matrigel coating is not clear to me.
2) Figure 2A and Figure 3A-3D : the y-axis legend seems to be missing. It will be self explanatory if authors could include it in figure.

Validity of the findings

The authors has shown and justified the rationality of the manuscript.

---

## Round 0.2 · accepted · Accept

Authors have addressed all concerns and now it meets the publication standard.
I noticed reviewer Dr. Zhang is unresponsive and a final editorial decision was made from my side.